# Telomere length as a biomarker for fetal fraction prediction in non-invasive prenatal testing

Zuzana Holesova[1]*, Jaroslav Budis[1,2,3], Marcel Kucharik[1,2,4], Juraj Gazdarica[1,2,3], Daria Carska[1], Gabriel Minarik[5,6], Michaela Hyblova[5,6], Tomas Szemes[1,2,7]

1 Geneton Ltd, Bratislava, Slovakia, 2 Comenius University Science Park, Bratislava, Slovakia, 3 Slovak Centre of Scientific and Technical Information, Bratislava, Slovakia, 4 Biomedical Research Center, Slovak Academy of Sciences, Bratislava, Slovakia, 5 TRISOMYtest Ltd, Nitra, Slovakia, 6 Medirex Group Academy, Nitra, Slovakia, 7 Department of Molecular Biology, Faculty of Natural Sciences, Comenius University, Bratislava, Slovakia

* zuzana.holesova@geneton.sk

## Abstract

Non-invasive prenatal testing (NIPT) has revolutionized prenatal diagnostics by providing a safer alternative to invasive techniques such as amniocentesis and chorionic villus sampling. NIPT detects chromosomal abnormalities through the analysis of cell-free fetal DNA (cffDNA) in maternal plasma. One of the critical factors influencing accuracy of NIPT is the fetal fraction (FF) – the proportion of fetal cell-free DNA relative to total cell-free DNA in maternal plasma. This study investigates the potential of using telomere length measurements as a novel biomarker for fetal fraction prediction in NIPT. Telomere-derived fragments, which differ between maternal and fetal DNA, may serve as a measure of FF due to the distinct telomere length. Specifically, deviations from the expected shorter telomere lengths of maternal DNA toward longer lengths could be more pronounced at higher FF levels. Various models incorporating telomere content and features selected by Ordinary Least Squares (OLS) were evaluated to enhance fetal fraction prediction. Our results showed that telomere content also works as an independent predictor (with Pearson correlation 0.23), yielding a small improvement in prediction precision when combined with traditional models.

## Introduction

Non-invasive prenatal testing (NIPT) has emerged as a pivotal tool in prenatal diagnostics, offering a safer alternative to invasive procedures like amniocentesis and chorionic villus sampling [1,2]. By analyzing cell-free fetal DNA (cffDNA) circulating in maternal plasma, NIPT enables the detection of fetal chromosomal abnormalities using several approaches with high sensitivity and specificity [3–5]. Since maternal and fetal cell-free DNA, which originates mainly from apoptotic placental trophoblasts, are mixed in the maternal bloodstream, one of the key challenges in NIPT is the accurate determination of the fetal fraction (FF), the proportion of cell-free fetal

**Data availability statement:** All relevant data are within the manuscript and its Supporting Information files.

**Funding:** Funded by the EU NextGenerationEU through the Recovery and Resilience Plan for Slovakia under the project No. 09I03-03-V03-00046. The funder provided support in the form of salaries for authors [JG], but did not have any additional role in the study design, data collection and analysis, decision to publish, or preparation of the manuscript. The specific roles of these authors are articulated in the 'author contributions' section.

**Competing interests:** We declare a potential competing financial interest in the form of employee contracts. ZH, JB, MK, JG, DC, and TS are the employees of Geneton Ltd., which participated in the development of the commercial NIPT test in Slovakia. MH and GM are the employees of Trisomy test Ltd. The authors declare no other conflict of interest. This does not alter our adherence to PLOS ONE policies on sharing data and materials.

DNA relative to the total cell-free DNA in maternal plasma. Accurate estimation of FF is essential, as low fetal fractions can lead to false-negative results or test failures, thereby impacting clinical decision-making [6,7].

Current methods for estimating FF primarily rely on detecting genetic or epigenetic variations that distinguish fetal DNA from maternal DNA, such as single nucleotide polymorphisms (SNPs), exploiting differences in DNA methylation patterns, quantifying Y-chromosome sequences in pregnancies with male fetuses, or *seqFF* method, which estimates FF based on fragment length profiles [8–13]. To enhance accuracy and overcome certain limitations of these approaches, we developed a complementary method in our laboratory, *comboFF*. This method builds on *SeqFF* by combining its length-based analysis with fragment count information from fetal-enriched regions along with attributes such as DNA library concentration, body mass index, and gestational age [13,14]. By integrating these additional features, *comboFF* provides more robust FF estimation, particularly in cases with low FF, while remaining independent of fetal sex and parental genotype data. These methods are effective, but they often require extensive sequencing, which can be cost-prohibitive or complex for some applications. A key challenge in FF measurement is variability due to biological and technical factors, such as maternal BMI, gestational age, and sequencing depth, which can introduce noise and impact accuracy. Additionally, current FF estimation approaches may be less reliable in pregnancies with female fetuses or cases involving chromosomal abnormalities.

Therefore, this study introduces telomere length as a potential complementary biomarker for FF estimation, offering an additional layer of insight that could enhance the accuracy of established FF assessment methods. Telomeres, the repetitive nucleotide sequences (TTAGGG in humans) at the ends of chromosomes, serve as protective caps that prevent chromosomal degradation during cell division [15]. They are fundamental to maintaining genomic stability and play a crucial role in cellular aging and senescence [16,17]. Due to the 'end-replication problem', a portion of telomeric DNA is lost with each cell division. Critically short telomeres lead to cell cycle arrest and apoptosis [18], limiting the proliferation of somatic cells. In contrast, germ and stem cells maintain telomeres via the action of enzyme telomerase, which is capable of elongating telomeres [19].

During fetal development, telomere length is known to undergo dynamic changes. It is typically longest in early embryonic cells and progressively shortens as the fetus matures. This phenomenon is thought to be related to the high proliferative capacity of embryonic and fetal cells, which require robust telomere maintenance mechanisms to support rapid cell division [20]. After birth, telomere attrition continues, influenced by factors such as age, lifestyle choices, and environmental exposures [21].

In the context of prenatal testing, the length of telomeres could provide valuable information about the proportion of fetal DNA present in a maternal blood sample. Since telomeres are longer in younger fetal cells than in maternal cells [22], we hypothesise that detecting the discrepancies in telomere content between fetal and maternal genomes could serve as a novel or complementary method for estimating

FF. As telomeres are influenced by cellular turnover and apoptosis, they could provide additional insights into FF beyond traditional genetic-based methods.

## Materials and methods

### Study design, sample preparation and sequencing

Maternal plasma samples were collected from pregnant women undergoing routine NIPT. Blood from pregnant women was collected. Blood plasma was separated and cell-free DNA was extracted. Standard fragment libraries for massively parallel sequencing were prepared using Illumina TruSeq Nano Kit sequenced with NextSeq 500/550 High Output Kits (San Diego, CA, USA). The study included various maternal factors such as age and gestational age to assess their potential influence on telomere content and FF estimation. The data were accessed for research purposes on 19 January 2024. The authors did not have access to information that could directly identify individual participants during or after data collection.

### Mapping of sequenced reads

Sequenced FASTQ files were mapped to human genome reference (hg19), using *Bowtie2* (v2.1.0) [23] resulting in one *Sequence Alignment Map* (SAM) file for each sample. SAM files were converted to *Binary Alignment Map* (BAM) format, sorted, and indexed using *Samtools* view, sort, and index utilities (v0.1.19) [24].

### Telomere analysis for prediction of fetal fraction in samples

In this study, we analyzed 5,073 maternal plasma samples from pregnancies with male fetuses to predict fetal fraction (FF) using telomere content. Male fetuses were chosen because the Y chromosome provides a reliable and unique marker for FF estimation, as it is absent in maternal DNA. This approach ensures accuracy, as FF was determined by measuring the abundance of Y chromosome fragments [9]. To assess telomere content, we employed the *TelomereHunter* tool [25], which estimates telomere content from whole-genome sequencing (WGS) data. Specifically, *TelomereHunter* calculates telomere content as the number of intratelomeric reads per million reads with telomeric GC content. In addition to calculating telomere content, *TelomereHunter* also estimates various other telomere characteristics, such as abundance of variant repeats in telomeric sequences (S1 Table). These measurements provide a more comprehensive profile of telomere behavior, which may improve the accuracy of fetal fraction predictions when combined with existing methods like *seqFF* [13] and *comboFF* [14].

### Data analysis

The following criteria were applied to filter samples for analysis. Only male samples were included, and the karyotype of the samples had to be XY. Only samples with a low risk result were included. The fetal fraction of the Y chromosome had to be less than 0.4, and samples with a final library concentration greater than 3 nM were selected. Additionally, the DNA quantity had to be between 0 and 4 ng/µL. Maternal age was restricted to the range of 15-55 years, and the gestational age of the fetus had to be between 50 and 200 days. These criteria ensured that only relevant and high-quality samples were included in the analysis.

### Model development and evaluation

To evaluate the predictive power of telomere-related features for fetal fraction (FF), a linear regression model was developed. The dataset was split into a training (80%, 4,058 samples) and a testing set (20%, 1,015 samples). The models were trained using *LinearRegression* from the *scikit-learn* Python library [26]. Ordinary Least Squares (OLS) regression module from library *statsmodels* [27] was used to select additional features, such as DNA quantity, gestational age, etc., and other telomere-related features from *TelomereHunter*, to optimize the model. The dependent variable (FF) was

derived from chromosome Y abundance, while independent variables included telomere content, other *TelomereHunter* features, other OLS-selected features, and *seqFF/comboFF* scores. Various model configurations were created to assess the impact of independent variables on FF prediction (schematically shown in Fig 1):

- **Model 1:** Telomere content

- **Model 2:** Telomere content + *seqFF/comboFF* model

- **Model 3:** *TelomereHunter*

- **Model 4:** *TelomereHunter* + *seqFF/comboFF* model

- **Model 5:** Telomere content + OLS-selected features

    ○ Alternative 1: *TelomereHunter* features only (S2 Table: features 4–14)

    ○ Alternative 2: *TelomereHunter* + other features (S2 Table: features 1–3)

- **Model 6:** Telomere content + OLS-selected features + *seqFF/comboFF* model

    ○ Alternative 1: *TelomereHunter* features only (S2 Table: features 4–14)

    ○ Alternative 2: *TelomereHunter* + other features (S2 Table: features 1–3)

| Model | Telomere content | Telomere Hunter | OLS Telomere Hunter | OLS others | seqFF | comboFF |
|-------|-----------------|-----------------|---------------------|------------|-------|---------|
| 1 | ✓ | | | | | |
| 2a | ✓ | | | | ✓ | |
| 2b | ✓ | | | | | ✓ |
| 3 | | ✓ | | | | |
| 4a | | ✓ | | | ✓ | |
| 4b | | ✓ | | | | ✓ |
| 5a | ✓ | | ✓ | | | |
| 5b | ✓ | | ✓ | ✓ | | |
| 6a | ✓ | | ✓ | | ✓ | |
| 6b | ✓ | | ✓ | | | ✓ |
| 6c | ✓ | | ✓ | ✓ | ✓ | |
| 6d | ✓ | | ✓ | ✓ | | ✓ |
| 7a | | | | | ✓ | |
| 7b | | | | | | ✓ |

**Fig 1. Model configurations used to assess the impact of telomere content, *TelomereHunter* features, and *seqFF/comboFF* models on fetal fraction (FF) prediction.** Each model being trained to predict FF both with and without maternal age as a feature to evaluate its influence on predictive performance.

- **Model 7:** *seqFF*/*comboFF* model only (baseline models)

Each model was trained to predict fetal fraction both with and without maternal age as a feature to evaluate the influence of maternal age on the predictive performance.

### Analysis aim

The primary objective of this analysis was to enhance the predictive accuracy of fetal fraction estimation by integrating telomere content into the existing *seqFF* and *comboFF* model. By examining multiple configurations, we aimed to determine how the inclusion of telomere content, maternal age, and OLS-selected features (selected from *TelomereHunter* and other features) would affect the model's performance. Through this comparative analysis, we sought to assess whether telomere content provides significant additional value for FF prediction beyond traditional genetic-based models.

### Statistical analysis

Pearson correlation coefficients and p-values were calculated for each model configuration to assess the predictive power of telomere content and the combined models. We used Python libraries: *NumPy* for efficient numerical computations [28], *SciPy* for statistical calculations [29], and *pandas* for data manipulation [30]. Plotting and visualization were performed with *matplotlib* and *seaborn* [31,32]. The performance of each model was evaluated based on the strength of correlation with the actual FF and statistical significance.

### Ethics approval and consent to participate

All the enrolled participants gave written informed consent for inclusion in this study. The study was conducted following the Declaration of Helsinki, and the protocol was approved by the Ethics Committee of the Bratislava Self-Governing Region on 30 June 2015 (03899/2015/HF), 25 March 2020 (05006/2020/HF/2) and 17 January 2023 (4530/2023/HF).

## Results

### Relationship between telomere content and conventional NIPT metrics

Across the data, telomere content showed a modest positive correlation with FF values derived from Y-chromosome, *seqFF*, and *comboFF* estimates, suggesting telomere content's potential for refining FF predictions (Fig 2).

Maternal age correlated weakly with slight declines in both telomere content and FF, while gestational age had minimal influence (Fig 3). The mean telomere content was 1559.6, with a median of 1517.9. The mean age of the mothers was 33.64 years, with a median of 34 years. The mean gestational age of the fetuses was 101.35 days, with a median gestational age of 96 days.

Technical factors such as higher DNA input and library concentration were linked to lower telomere content, highlighting the need to account for these in model calibration (Fig 4). Together, these results provide foundational insights for integrating telomere and FF metrics in predictive models.

For FF estimations using Y-chromosome, *seqFF*, and *comboFF* methods, a slight decrease in FF was observed with increasing maternal age across all methods (Figs 5A, B, C). Conversely, FF values estimated using the Y-chromosome and *seqFF* methods showed a marginal increase with gestational age, but these results were not statistically significant (Figs 5D, E). The *comboFF* method exhibited a slight decrease in FF with gestational age, but this correlation was also non-significant (Fig 5F).

### Fetal fraction prediction

The results from the Ordinary Least Squares (OLS) analysis in (S2 Table) highlight key variables significantly associated with telomere content. First, the experimental inputs of DNA quantity and final library concentration have negative

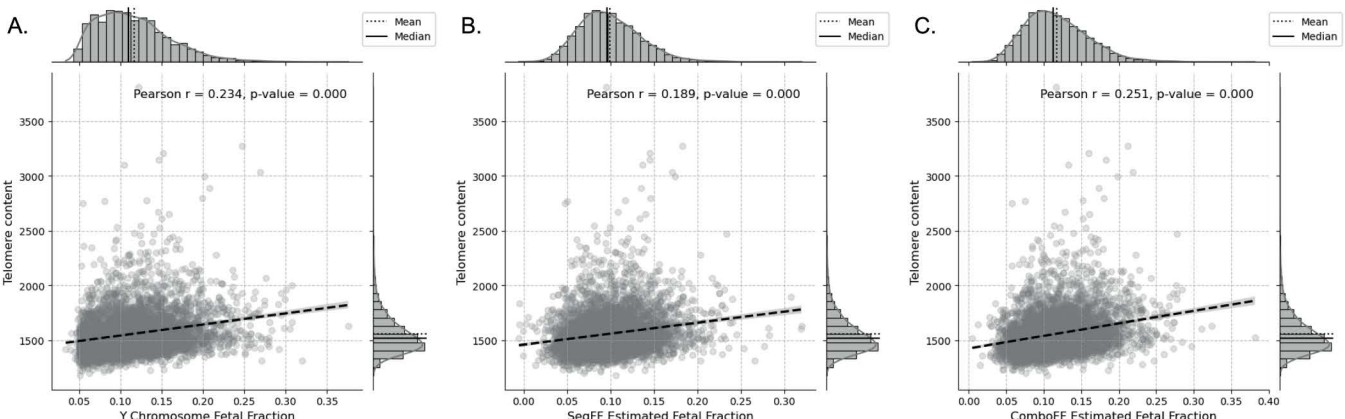

**Fig 2. Effect of fetal fraction estimated using Y chromosome, *seqFF*, and *comboFF* methods on the telomere content.** Telomere content shows a moderate increase with higher fetal fraction estimated using the Y chromosome method, indicating a positive correlation (Pearson's r = 0.234, p < 1.00e-05 **(A)**. A similar trend is seen with the *seqFF* method, where telomere content increases with fetal fraction, though slightly less strongly (Pearson's r = 0.189, p < 1.00e-05) **(B)**. The *comboFF* method demonstrates the strongest association, with telomere content rising in line with fetal fraction (Pearson's r = 0.251, p < 1.00e-05) **(C)**. This positive relationship between fetal fraction and telomere content motivated the inclusion of telomere metrics in predictive models for FF estimation, hypothesizing that telomere-based data could enhance FF prediction accuracy.

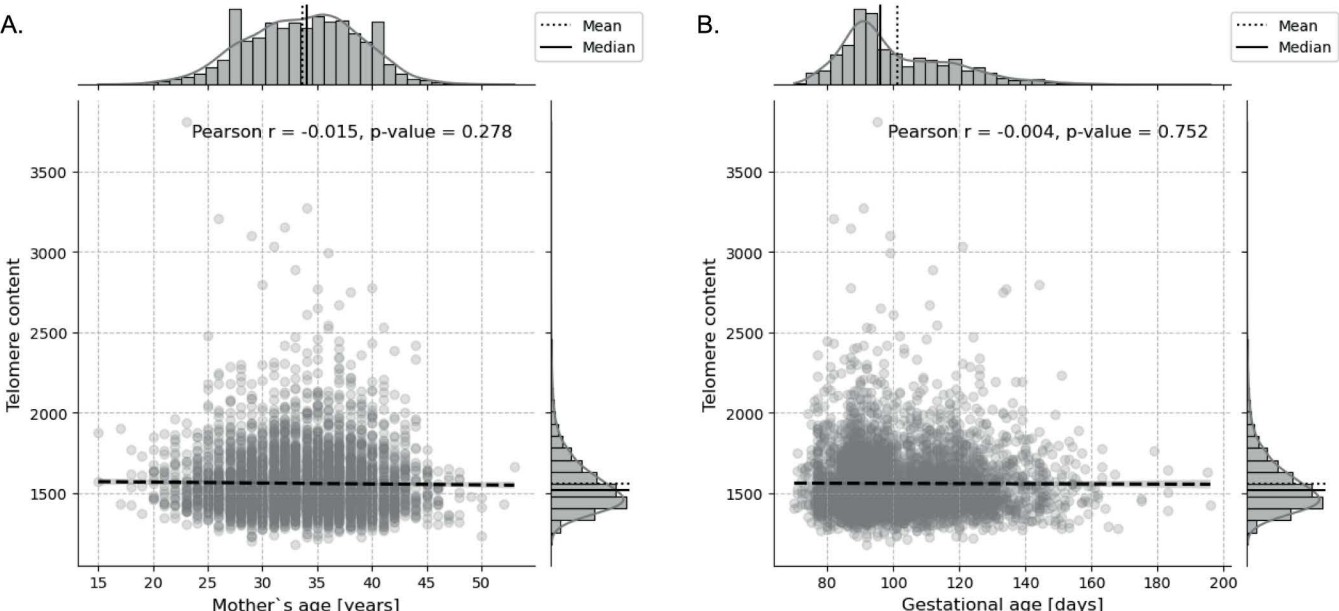

**Fig 3. The effect of maternal and gestational age on telomere content.** A very slight decrease in telomere content with increasing maternal age, but the association is not statistically significant (Pearson's r = −0.015, p = 0.278) **(A)**. Similarly, telomere content shows a minor decline with increasing gestational age, with no significant correlation (Pearson's r = −0.004, p = 0.752) **(B)**. Although telomere content showed a downward trend with increasing gestational age as expected, the effect was very weak. Maternal age did not show a correlation with telomere content.

coefficients, indicating that telomere content tends to decrease as these values increase. These effects are statistically significant, with very low p-values (1.00e-05 and 3.42e-28, respectively), underscoring their influence on telomere content. Additionally, gestational age exhibits a small positive effect (coefficient 2.98e-05, p = 1.72e-02), suggesting that telomere content may slightly increase as gestational age progresses, though the effect size is minimal.

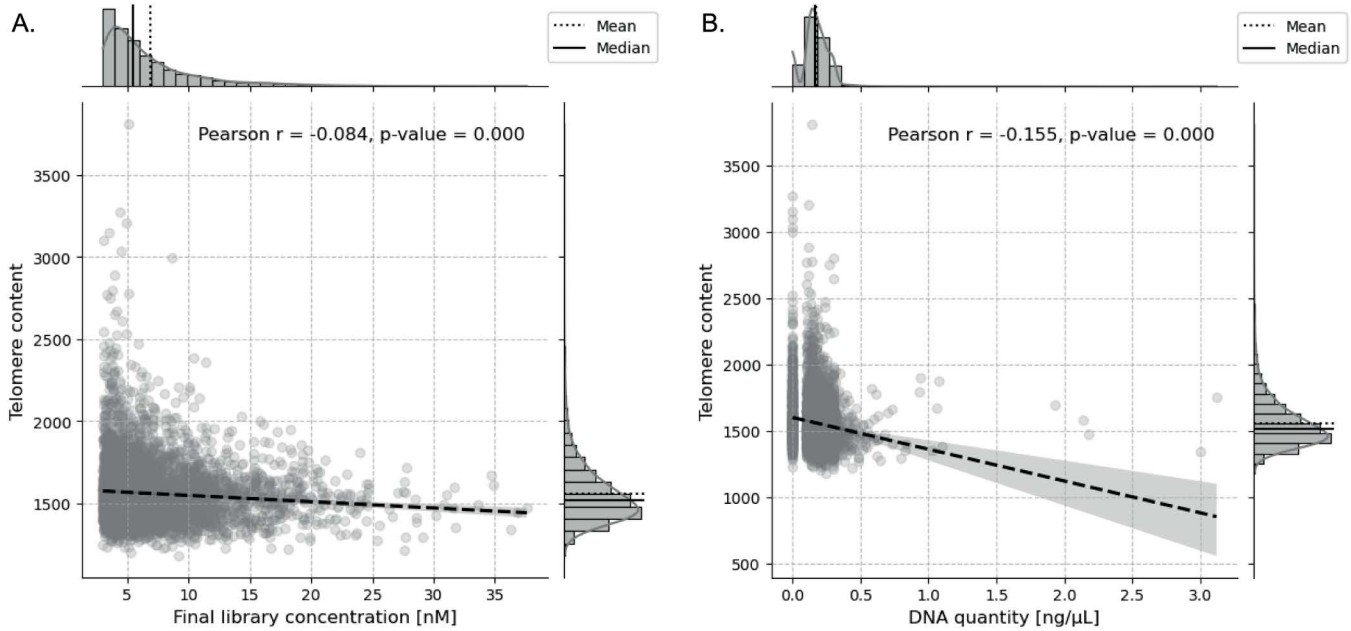

**Fig 4. Effect of final library concentration and DNA quantity on telomere content.** Telomere content shows a slight decrease as final library concentration increases, with a small but statistically significant negative correlation (Pearson's $r = -0.084$, $p < 1.00e-05$) **(A)**. A stronger negative relationship exists between telomere content and DNA quantity, with telomere content decreasing as DNA quantity increases (Pearson's $r = -0.155$, $p < 1.00e-05$), indicating that higher DNA quantities may affect telomere measurements more substantially **(B)**.

The analysis also reveals that specific features derived from *TelomereHunter* have a pronounced impact. For example, *TTAGGG_norm_by_intratel* has the highest positive coefficient (0.1338) and an extremely low p-value (2.15e-105), indicating a strong relationship with telomere content. Conversely, other features like *TCAGGG_norm_by_intratel* and *TGAGGG_norm_by_intratel* show negative coefficients with highly significant p-values (2.25e-07 and 8.68e-08, respectively), pointing to their potential inverse effect on telomere content.

## Comparative analysis of FF estimation models

The comparative performance of various models for fetal fraction (FF) estimation, as detailed in Table 1, reveals several key insights into the predictive power of different approaches.

### Telomere content and other *TelomereHunter* features as a predictor of fetal fraction

Basic models using only telomere content, with or without maternal age, exhibited low correlation with FF ($r = 0.2338 / 0.2365$) and relatively high RMSE (0.0435/ 0.0434), indicating limited accuracy (Fig 6). In contrast, when *TelomereHunter* was applied with telomere content, the correlation improved significantly ($r = 0.4598$), and RMSE dropped to 0.0397, demonstrating moderate predictive improvement over simpler telomere-based models (S1 Fig).

Models combining *seqFF* or *comboFF* with telomere content yielded substantially better performance (Figs 6B, E). Specifically, *seqFF* paired with telomere content (Model 7a) achieved a strong correlation ($r = 0.8751$) and a reduced RMSE of 0.0216. When maternal age was also included, *comboFF* (Model 7b) achieved one of the lowest RMSE values (0.0190) and maintained a high correlation of $r = 0.9057$, underscoring the effectiveness of combining these methodologies.

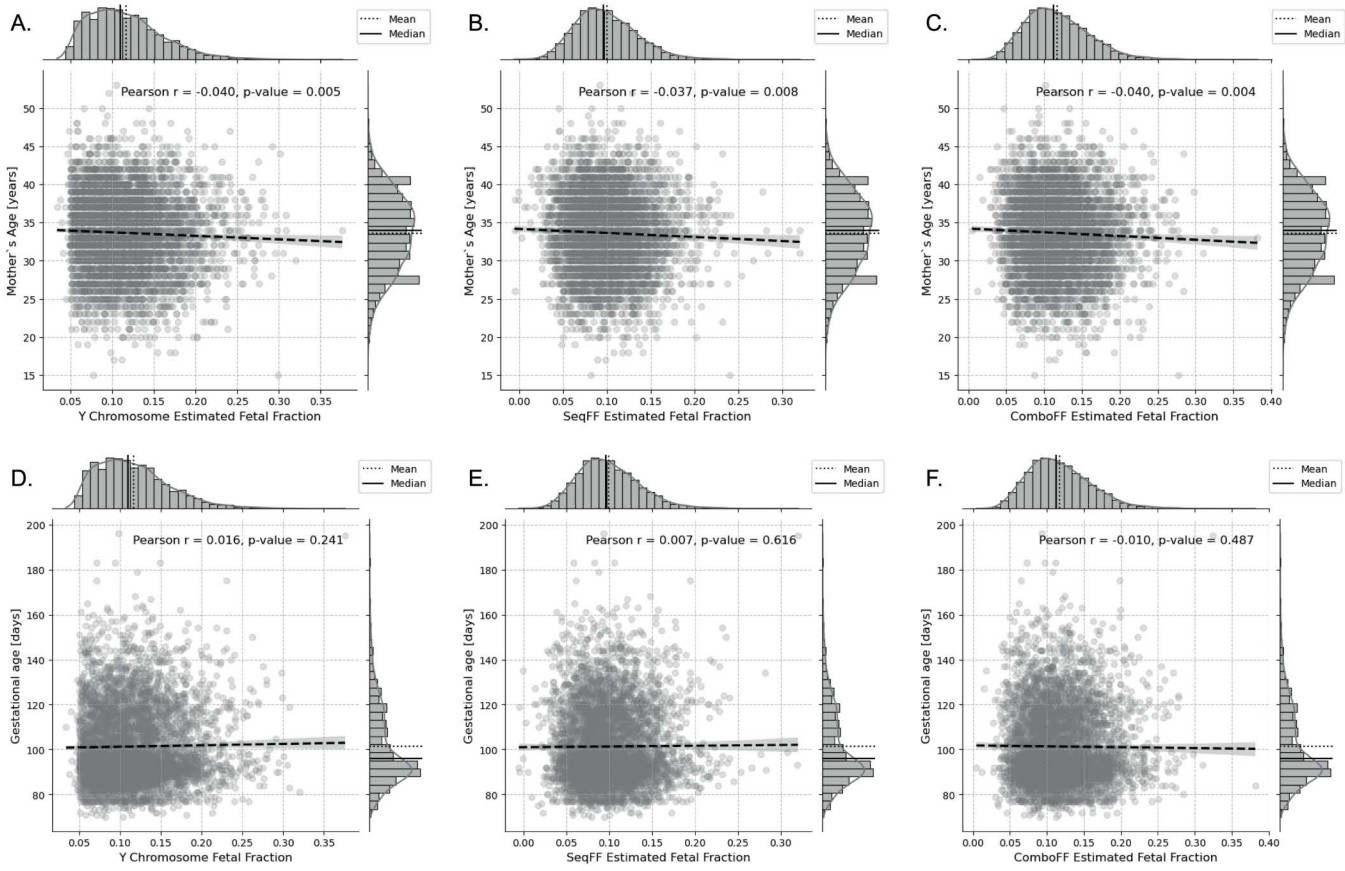

**Fig 5. Effect of maternal and gestational age on the fetal fraction estimated using Y-chromosome, *seqFF* and *comboFF* methods.** There is a slight decrease in fetal fraction with increasing maternal age for all methods (Pearson's r = −0.040/-0.037/-0.040, p = 0.005/0.008/0.004) **(A, B, C)**. In contrast, fetal fraction estimates using Y-chromosome and *seqFF* method exhibit a slight increase with gestational age, though not statistically significant (Pearson's r = 0.016/0.007, p = 0.241/0.616) **(D, E)**. The *comboFF* method shows a slight decrease in fetal fraction with increasing gestational age (Pearson's r = 0.251, p = 0.487), but this also lacks significance **(C)**.

Adding *TelomereHunter* features to *seqFF* and *comboFF* further enhanced predictive accuracy. For instance, Model 4b, which utilized *comboFF* with *TelomereHunter* features, achieved an excellent correlation of r = 0.9064 and the lowest RMSE of 0.0189 among these combinations (S1 Fig).

### Impact of OLS-selected features from TelomereHunter and other sources

The incorporation of OLS-selected features from *TelomereHunter*, along with data from other sources, significantly enhanced model performance. Specifically, these features contributed to improved correlation coefficients and reduced root mean square error (RMSE) values, indicating a better fit between predicted and actual FF values. For instance, Model 5a (S2 Fig), which integrated OLS-selected features with telomere content, achieved a moderate correlation (r = 0.45), contrasting with the lower correlation of telomere content alone (Model 1, r = 0.23). This improvement highlights the value of targeted features in increasing predictive accuracy.

Moreover, the top-performing configurations, particularly Model 6d (Fig 7), which combined OLS-selected features with telomere content, maternal age, and the *comboFF* method, reached the highest correlation (r = 0.9072) and the lowest RMSE (0.0189) among all tested models. This performance underscores the significance of OLS-selected features from

**Table 1. Comparative performance of models for fetal fraction estimation using telomere content, *TelomereHunter* features, *seqFF*, and *comboFF* methods.**

| Model | Description | Pearson correlation coefficient (r) (without maternal age) | Pearson correlation coefficient (r) (with maternal age) | RMSE | p-value |
|---|---|---|---|---|---|
| **1** | Telomere content /+maternal age | 0.2338 | **0.2365** | 0.0435/0.0434 | 6.381e-64/1.985e-65 |
| **2a** | Telomere content + *seqFF* model /+maternal age | 0.8751 | 0.8751 | 0.0216/0.0216 | <1.00e-05/<1.00e-05 |
| **2b** | Telomere content + *comboFF* model /+maternal age | 0.9057 | 0.9057 | 0.0190/0.0190 | <1.00e-05/<1.00e-05 |
| **3** | *TelomereHunter* /+maternal age | 0.4598 | 0.4598 | 0.0397/0.0397 | 8.1e-264/7.438e-264 |
| **4a** | *TelomereHunter* + *seqFF* model /+maternal age | 0.8784 | 0.8784 | 0.0214/0.0214 | <1.00e-05/<1.00e-05 |
| **4b** | *TelomereHunter* + *comboFF* model /+maternal age | 0.9064 | 0.9064 | 0.0189/0.0189 | <1.00e-05/<1.00e-05 |
| **5a** | Telomere content + OLS-selected features (*TelomereHunter* only) /+maternal age | 0.4490 | 0.4490 | 0.0399/0.0399 | 3.288e-250/5.206e-254 |
| **5b** | Telomere content + OLS-selected features (*TelomereHunter* combined with others) /+maternal age | 0.4848 | **0.4849** | 0.0391/0.0391 | 1.886e-297/1.793e-297 |
| **6a** | Telomere content + OLS-selected features (*TelomereHunter* only) + *seqFF* model /+maternal age | 0.8783 | 0.8783 | 0.0214/0.0214 | <1.00e-05/<1.00e-05 |
| **6b** | Telomere content + OLS-selected features (*TelomereHunter* only) + *comboFF* /+maternal age | 0.9063 | 0.9063 | 0.0189/0.0189 | <1.00e-05/<1.00e-05 |
| **6c** | Telomere content + OLS-selected features (*TelomereHunter* combined with others) + *seqFF* /+maternal age | 0.8806 | 0.8806 | 0.0212/0.0212 | <1.00e-05/<1.00e-05 |
| **6d** | Telomere content + OLS-selected features (*TelomereHunter* combined with others) + *comboFF* /+maternal age | 0.9072 | 0.9072 | 0.0189/0.0189 | <1.00e-05/<1.00e-05 |
| **7a** | *seqFF* model only (baseline model) /+maternal age | 0.8723 | 0.8723 | 0.0278/0.0278 | <1.00e-05/<1.00e-05 |
| **7b** | *comboFF* model only (baseline model) /+maternal age | 0.9056 | 0.9056 | 0.0191/0.0191 | <1.00e-05/<1.00e-05 |

**Pearson correlation coefficient (r)** - measures the strength and direction of the relationship between two variables; **root mean square error (RMSE)** - measures the average difference between a statistical model's predicted values and the actual values; **p-value** - the strength of evidence against the null hypothesis, indicating whether the observed data could occur by chance (p-value < 0.05 suggests that the coefficient is statistically significant, implying a meaningful association between the variable and the response);**/+maternal age** - represent results obtained after adding maternal age to the model, where values before and after the slash indicate results without and with maternal age included, respectively.

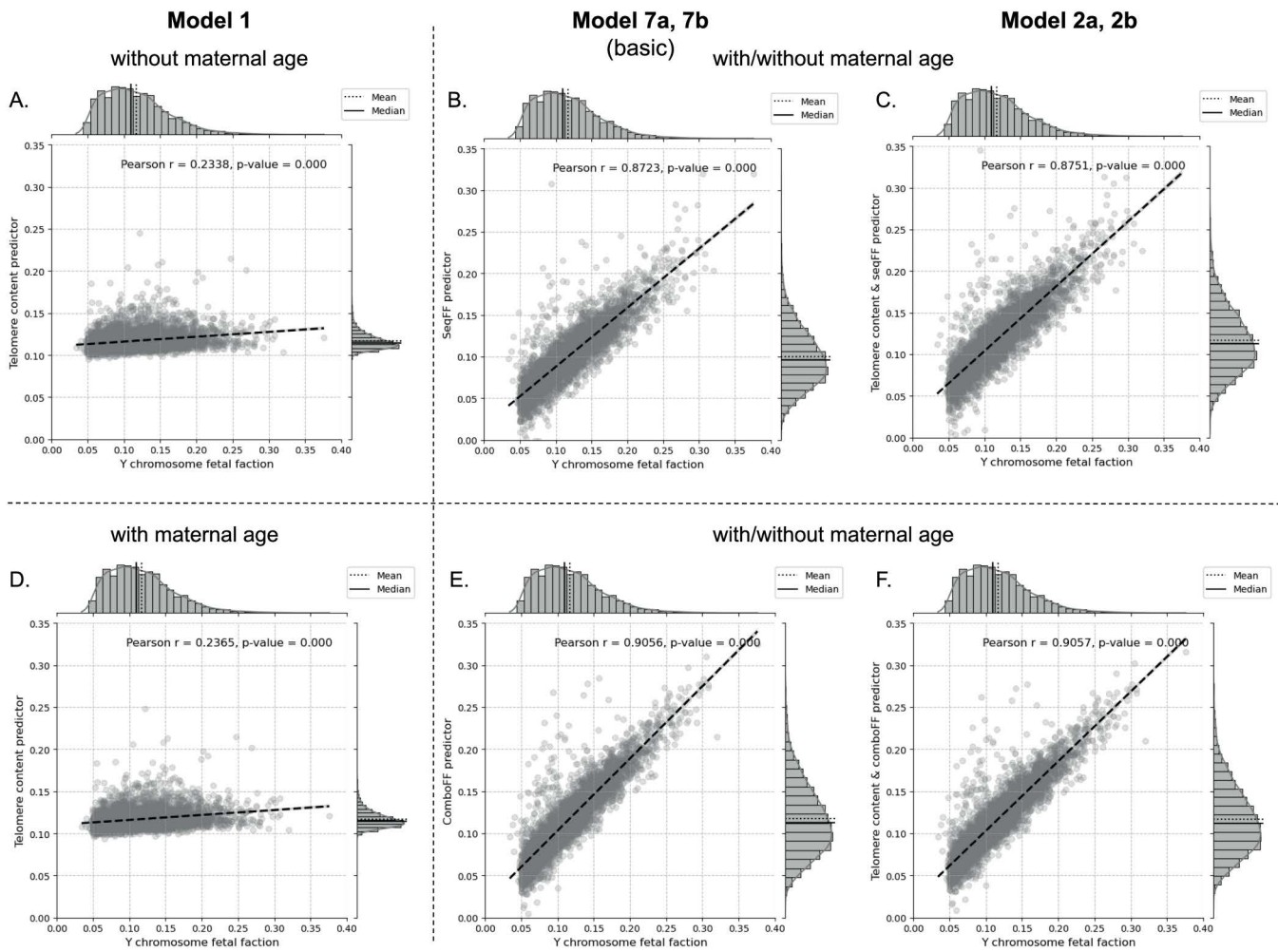

**Fig 6. Comparison of the effectiveness of Models 1, 2a and 2b in estimating fetal fraction (FF) based on telomere content.** Model 1, which uses telomere content with (A) or without maternal age **(D)**, demonstrates low correlation with FF (r = 0.23) and high RMSE (0.043), indicating limited predictive accuracy. The p-values were statistically significant (6.381e-64/1.985e-65). When *seqFF* was combined with telomere content in Models 2a (C) and 2b **(F)**, the correlation increased substantially (r = 0.8751 for both). The RMSE decreased to 0.0216 for the *seqFF* model and 0.0190 for the *comboFF* model, indicating a strong predictive improvement. Incorporating maternal age did not further enhance correlation but maintained statistical significance (p-value < 1.00e-05 for all comparisons). Models utilizing *seqFF* alone (Model 7a **(B)**) and *comboFF* alone (Model 7b **(E)**) achieved high correlations (r = 0.8723 and r = 0.9056, respectively), demonstrating the baseline predictive power of these models without additional features.

*TelomereHunter*, which likely capture essential variations that enhance FF estimation. The motivation behind including these features was rooted in the necessity for more precise FF predictions in prenatal screening contexts, where accuracy is paramount. Overall, the integration of OLS-selected features, especially when paired with advanced predictive methods, demonstrates a substantial advantage in achieving reliable FF estimation models.

## Best model performance

Model 6d (Fig 7), which combines *comboFF* with telomere content and OLS-selected features from *TelomereHunter* (including maternal age), demonstrated the best performance for estimating fetal fraction (FF). It achieved the highest Pearson correlation coefficient (r = 0.9072, p < 1.00e-05), indicating a strong association between predicted and actual FF

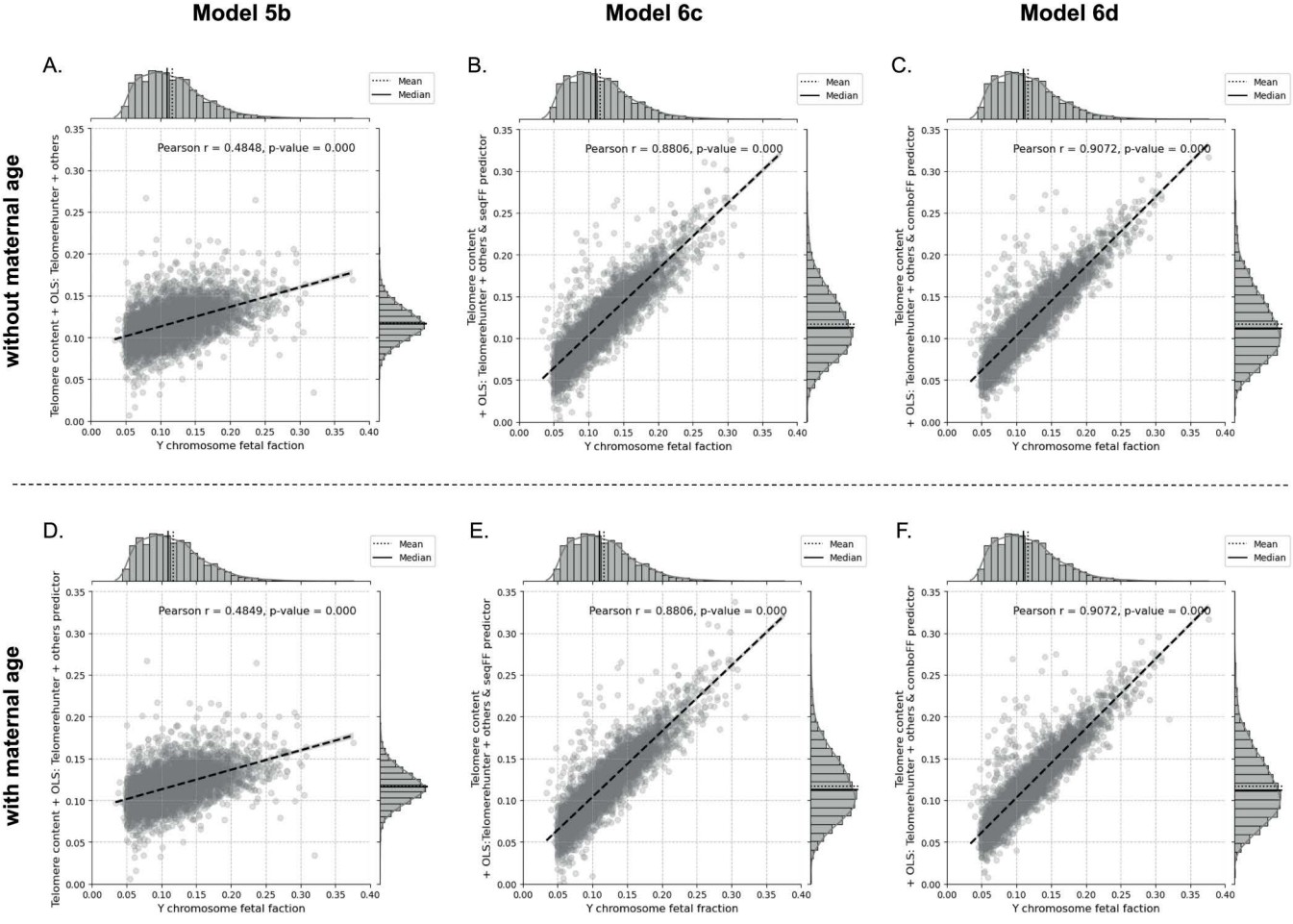

**Fig 7. Comparison of the effectiveness of Models 5b, 6c, and 6d in estimating fetal fraction (FF) using OLS-selected features from *Telomere-Hunter* and other sources.** Model 5b, tested with (A) and without (D) maternal age, demonstrates a moderate improvement in accuracy, achieving correlation values (r = 0.4848/0.4849) and RMSE (0.0391), showing a step up from models based solely on telomere content. Model 6c **(B, E)**, which integrates *seqFF*, yields a substantial boost in correlation (r = 0.8806) and a lower RMSE of 0.0212, highlighting *seqFF*'s enhanced effectiveness with additional features. Model 6d **(C, F)**, which combines *comboFF*, reaches the highest accuracy among the tested models, achieving the strongest correlation (r = 0.9072) and the lowest RMSE (0.0189), marking it as the most effective configuration for FF estimation in this analysis.

values, and recorded the lowest RMSE (0.0189), highlighting minimal error in predictions. These values underscore Model 6 d's accuracy and effectiveness, setting it apart as the most reliable configuration among all tested models in the study.

## Effect of maternal age

In the context of fetal fraction (FF) prediction models, maternal age had a relatively small impact. When maternal age was included as a variable, it generally didn't significantly change the correlation or RMSE values for most models. For example, in Model 1 (which included telomere content alone), the addition of maternal age led to only slight improvement in predictive accuracy (from r = 0.2338 to r = 0.2365). The best-performing model, Model 6d, maintained its high correlation (r = 0.9072) and lowest RMSE (0.0189) with or without maternal age, suggesting that while age was accounted for, it wasn't a major driver of predictive improvement in this model. This limited influence may be attributed to the relatively narrow age range of the mothers, most of whom were of similar age, reducing the variability in maternal age as a predictive factor.

## Discussion

This study investigated the potential of telomere content as an additional biomarker for improving fetal fraction (FF) estimation in non-invasive prenatal testing (NIPT). Incorporating telomere content and associated *TelomereHunter* features into models showed modest improvements of traditional methods like *seqFF* and *comboFF*. This suggests that while there may be detectable telomere content differences between maternal and fetal DNA, these variations may contribute to FF prediction when compared to established genetic-based methods.

Telomere content, often linked to cellular aging and genetic stability, has recently been explored for its relevance in prenatal diagnostics, as telomere integrity can reflect fetal and maternal cellular dynamics [33]. Prior research indicates that shorter telomeres may correlate with adverse pregnancy outcomes, emphasizing telomeres' potential utility in prenatal care. This study demonstrated that combining telomere features with other predictors only modestly improved FF prediction, with the best model (Model 6d) achieving a strong correlation (r = 0.9072) and low RMSE (0.0189).

Maternal age showed only a weak association with FF, which is consistent with prior studies suggesting that its apparent effect may stem from confounding factors like maternal BMI. Higher maternal BMI has been strongly correlated with lower FF levels [34]. Similarly, when maternal age was combined with telomere content and other features, no substantial improvement in predictive performance was observed, reinforcing the limited influence of age alone on FF prediction. This aligns with recent evidence suggesting that age-related telomere dynamics, though notable, may not substantially influence FF predictions [7]. The observed slight negative correlation between maternal age and FF (r = −0.040) indicates that while telomere biology might offer additional insights, its standalone effect may not surpass other genetic markers in predictive strength. Nevertheless, the modest improvement observed in our study indicates that telomere-based predictions could enhance multiparametric models. Overall, this study highlights the potential of telomere-based features in FF prediction while emphasizing the need for further research to refine their application in prenatal screening. Investigating how telomere dynamics vary across different pregnancies and integrating additional biomarkers could further improve FF estimation, particularly in complex clinical scenarios. Combining telomere content with established biomarkers may contribute to a more comprehensive approach for assessing fetal health, ultimately enhancing the accuracy and reliability of prenatal diagnostics.

## Limitations

Several factors should be considered when interpreting these findings. Both genetic and environmental influences play a role in determining telomere length [35,36]. Although there is no gender difference at birth, adult females typically have longer telomeres than males [37]. Additionally, telomere length varies across racial and ethnic groups [38,39]. Moreover, telomere length is influenced by a range of health conditions, including obesity [40], diabetes [41], some cancers [42], and cardiovascular disease [43]. Furthermore, external factors such as mental health [44,45], behaviors [46], and external environmental conditions [47] may contribute to telomere variability, potentially affecting its reliability as a biomarker in prenatal testing.

Additionally, this study focused on pregnancies with healthy male fetuses, using the Y-chromosome as a reference for fetal fraction (FF) estimation. While this approach minimizes certain confounding factors and allows for a straightforward FF measurement, it limits the generalizability of our findings to pregnancies with female fetuses, where alternative methods for fetal DNA discrimination would be required. Moreover, pregnancies affected by chromosomal abnormalities or other complications were not included, and thus, the applicability of telomere-based features in such cases remains unclear. These variables introduce additional complexity that future studies should address by examining larger and more diverse cohorts.

## Conclusion

In summary, this study highlights that while telomere content can contribute to estimating fetal fraction (FF) in non-invasive prenatal testing (NIPT), their impact remains limited compared to established genetic-based models such as

*seqFF* and *comboFF*. The best-performing model, which combined telomere metrics with selected features and maternal age, achieved the highest accuracy, indicating that a multifactorial approach may offer benefits. However, the modest improvement suggests that telomere length may serve as a complementary method for refining established FF prediction methods. Our findings also emphasize the value of integrating multiple biomarkers to refine FF prediction models. Although telomere content alone did not markedly enhance predictive accuracy of fetal fraction in our study, further research into telomere dynamics and their relationship with FF across diverse maternal and gestational age ranges could lead to more applications and shed more light on their potential role in prenatal diagnostics.

## Supporting information

**S1 Fig. Comparison of the effectiveness of Models 3, 4a, and 4b in estimating fetal fraction (FF) based on all *TelomereHunter* features, including telomere content.** Model 3 demonstrates a better correlation (r = 0.4598) and a slightly lower RMSE of 0.0397, indicating an improvement over the use of telomere content alone. This model was highly significant (p-values = 8.1e-264/7.438e-264). *SeqFF* with *TelomereHunter* features (Model 4a) achieved a correlation of r = 0.8784 and an RMSE of 0.0214. In contrast, *comboFF* with *TelomereHunter* features (Model 4b) reached an even higher correlation of r = 0.9064 and the lowest RMSE of 0.0189. For further details, refer to Models 7a and 7b in Fig 6. (TIF)

**S2 Fig. Comparison of the effectiveness of Models 5a, 6a, and 6b in estimating fetal fraction (FF) using OLS-selected features from *TelomereHunter*.** Model 5a, which incorporates OLS-selected features alongside telomere content, yields a moderate correlation (r = 0.4490/0.4490) and an RMSE of approximately 0.0399, indicating that while the inclusion of targeted features provides some improvement, it remains less effective than more comprehensive models. In contrast, Model 6a achieves a high correlation of r = 0.8783/0.8783 and an RMSE of 0.0214, affirming the effectiveness of *seqFF* with selected features. Model 6b reaches r = 0.9063/0.9063 with an RMSE of 0.0189, marking this combination as one of the top-performing configurations. (TIF)

**S1 Table. *TelomereHunter* features.** (DOCX)

**S2 Table. Ordinary least squares (OLS) regression results.** [1]quantity of input DNA used in the sequencing process, [2]concentration of the final prepared sequencing library, [3]gestational age refers to the age of the fetus in days, [4]total count of reads containing telomeric sequences, [5]counts intratelomeric reads or reads with variations in the standard telomeric repeat sequence, [6]total count of reads containing telomeric sequences, including GC-content adjustments, [7]telomere content in the sample, calculated as the proportion of reads containing telomeric repeats (e.g., TTAGGG sequences) relative to total reads, [8-14]normalized count of reads containing the exact TTAGGG/TCAGGG/TGAGGG/TTTGGG/GTTGGG/AGTGGG/AGAGGG repeat sequence divided by the number of intratelomeric reads. [4-14]OLS-selected features from *TelomereHunter*, [1-3]others OLS-selected features. (DOCX)

**S1 File. Raw experimental data.** (XLSX)

## Acknowledgments

We would like to thank all of the participating centers who submitted samples used in this study. Additionally, we'd like to thank the patients who agreed to participate. We appreciate the support of the laboratory staff for their contributions.

## Author contributions

**Conceptualization:** Jaroslav Budis, Juraj Gazdarica, Tomas Szemes.

**Data curation:** Marcel Kucharik, Gabriel Minarik, Michaela Hyblova.

**Formal analysis:** Zuzana Holesova.

**Funding acquisition:** Tomas Szemes.

**Methodology:** Jaroslav Budis, Marcel Kucharik, Daria Carska.

**Project administration:** Jaroslav Budis.

**Resources:** Gabriel Minarik, Michaela Hyblova, Tomas Szemes.

**Software:** Zuzana Holesova, Marcel Kucharik, Daria Carska.

**Supervision:** Jaroslav Budis.

**Validation:** Zuzana Holesova, Marcel Kucharik.

**Visualization:** Zuzana Holesova, Daria Carska.

**Writing – original draft:** Zuzana Holesova.

**Writing – review & editing:** Jaroslav Budis, Marcel Kucharik, Juraj Gazdarica.

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
