## [Decision Letter · Decision Letter 0]

PONE-D-24-57732Telomere length as a biomarker for fetal fraction prediction in non‑invasive prenatal testingPLOS ONE

Dear Dr. Holesova, 

Thank you for submitting your manuscript to PLOS ONE. After careful consideration, we feel that it has merit but does not fully meet PLOS ONE’s publication criteria as it currently stands. Therefore, we invite you to submit a revised version of the manuscript that addresses the points raised during the review process.

We look forward to receiving your revised manuscript.

Kind regards,

Giuseppe Novelli

Academic Editor

PLOS ONE

Journal Requirements:

“Funded by the EU NextGenerationEU through the Recovery and Resilience Plan for Slovakia under the project No. 09I03-03-V03-00046 and by the project FORGENOM II, funded by the Horizon Europe program under grant agreement No. 101160008.”

“We declare a potential competing financial interest in the form of employee contracts. ZH, JB, MK, JG, DC and TS are the employees of Geneton Ltd., which participated in the development of the commercial NIPT test in Slovakia. MH and GM are the employees of Trisomy test Ltd. The authors declare no other conflict of interest. This does not alter our adherence to PLOS ONE policies on sharing data and materials.”

We note that one or more of the authors are employed by a commercial company: Geneton Ltd., Trisomy test Ltd.

2) Please also provide an updated Competing Interests Statement declaring this commercial affiliation along with any other relevant declarations relating to employment, consultancy, patents, products in development, or marketed products, etc.  

Within your Competing Interests Statement, please confirm that this commercial affiliation does not alter your adherence to all PLOS ONE policies on sharing data and materials by including the following statement: ""This does not alter our adherence to  PLOS ONE policies on sharing data and materials.” (as detailed online in our guide for authors http://journals.plos.org/plosone/s/competing-interests). If this adherence statement is not accurate and  there are restrictions on sharing of data and/or materials, please state these. Please note that we cannot proceed with consideration of your article until this information has been declared.

5. Please ensure that you include a title page within your main document. You should list all authors and all affiliations as per our author instructions and clearly indicate the corresponding author.

7. We notice that your supplementary figures are uploaded with the file type 'Figure'. Please amend the file type to 'Supporting Information'. Please ensure that each Supporting Information file has a legend listed in the manuscript after the references list.

**Additional Editor Comments:**

Authors are encouraged to review their manuscript and integrate it with the reviewers' suggestions.

Reviewers' comments:

Reviewer's Responses to Questions

**Comments to the Author**

1. Is the manuscript technically sound, and do the data support the conclusions?

Reviewer #1: Yes

Reviewer #2: Yes

Reviewer #3: Yes

2. Has the statistical analysis been performed appropriately and rigorously? 

Reviewer #1: Yes

Reviewer #2: Yes

Reviewer #3: Yes

3. Have the authors made all data underlying the findings in their manuscript fully available?

Reviewer #1: No

Reviewer #2: Yes

Reviewer #3: Yes

4. Is the manuscript presented in an intelligible fashion and written in standard English?

Reviewer #1: Yes

Reviewer #2: Yes

Reviewer #3: Yes

5. Review Comments to the Author

Reviewer #1: The authors explore the hypothesis that telomere length could serve as a novel biomarker for predicting fetal fraction (FF) in non-invasive prenatal testing (NIPT). While the integration of telomere-related metrics into applied models (e.g., seqFF and comboFF) provides modest improvements, the overall impact appears limited compared to established genetic-based methodologies. Nevertheless, this research offers valuable insights into potential supplementary approaches for FF estimation. I would like to share some comments and suggestions to enhance the manuscript's clarity and rigor.

Major Comments:

-The correlation coefficients reported (e.g., telomere content alone vs. best-performing model) suggest that telomere metrics provide only incremental improvements when integrated with conventional tools. Consider elaborating on the practical implications of these modest gains (e.g., Would the improved prediction accuracy significantly impact clinical outcomes or test reliability?).

-The manuscript would benefit from a deeper insights of the biological variability of telomere length across populations (e.g., potential influences of maternal health, genetic background, or environmental factors on telomere dynamics). Furthermore, a discussion is necessary on whether the observed telomere-derived signals are consistent across diverse maternal-fetal conditions (e.g., healthy fetuses vs affected).

-Although the models incorporate telomere metrics and maternal age, the manuscript does not sufficiently address potential biases or limitations (e.g., the exclusion of pregnancies with female fetuses; technical variability in telomere measurement; the impact of maternal BMI or other confounding factors). Including a limitations section to discuss these points would strengthen the study.

-Indicate as supporting data will be shared to improve transparency and reproducibility.

Minor suggestions:

-Line 23: placental trophoblast is the main but not sole source of cffDNA. A lesser source is apoptosis of erythroblasts in the fetal circulation, which generates cfDNA that can cross the placenta and enter the maternal circulation (PMID: 11113889; 12200465).

-Ensure consistent use of terms like “telomere content,” “TelomereHunter features,” and “fetal fraction prediction models” throughout the manuscript.

-The Introduction is effective but could provide more context on the limitations of current FF estimation methods, highlight the potential gaps that telomere metrics aim to fill.

-In Discussion, emphasize how the findings align or diverge from prior research on telomere dynamics in prenatal diagnostics.

Reviewer #2: The accuracy of methods for quantifying fetal DNA fraction in maternal plasma samples during pregnancy is maximized by two main factors, fetal sex (M) and sequencing coverage (>10M reads). In the absence of these requirements, further criteria for defining FF require robust statistical processing dependent on data variability. The search for new models must overcome this last problem. This is not the case for telomere sequences which, although decreasing with age, do not easily distinguish placenta from maternal lymphocytes due to the innumerable inter and intra-individual variables. It is therefore not surprising that the authors did not draw favorable conclusions in their work, nor does it appear useful to share such data.

Reviewer #3: This study investigates the potential utility of telomere content as an additional biomarker to enhance the estimation of fetal fraction (FF) in non-invasive prenatal testing.

The manuscript is clearly written, methodologically sound, and well-structured.

The results suggest that telomere length may serve as a “complementary tool” for refining established FF prediction methods.

Although the inclusion of telomere-related features yielded only modest improvements in FF prediction accuracy, the study provides valuable insights and highlights the importance of integrating multiple biomarkers to advance FF estimation models.

6. PLOS authors have the option to publish the peer review history of their article (what does this mean? ). If published, this will include your full peer review and any attached files.

**Do you want your identity to be public for this peer review?** For information about this choice, including consent withdrawal, please see our Privacy Policy .

Reviewer #1: **Yes: ** Ludovico Graziani

Reviewer #2: No

Reviewer #3: No

---

## [Author Response · Author response to Decision Letter 1]

2 May 2025

Information regarding the Funding Statement and Competing Interests Statement has been included in the Cover letter (Statements) and Responses to the reviewers in the form of a Rebuttal letter.

---

## [Decision Letter · Decision Letter 1]

Telomere length as a biomarker for fetal fraction prediction in non‑invasive prenatal testing

PONE-D-24-57732R1

Dear Dr. Holesova,

We’re pleased to inform you that your manuscript has been judged scientifically suitable for publication and will be formally accepted for publication once it meets all outstanding technical requirements.

Kind regards,

Rishi Jaiswal, Ph.D.

Academic Editor

PLOS ONE

Additional Editor Comments (optional):

Reviewers' comments:

Reviewer's Responses to Questions

**Comments to the Author**

1. If the authors have adequately addressed your comments raised in a previous round of review and you feel that this manuscript is now acceptable for publication, you may indicate that here to bypass the “Comments to the Author” section, enter your conflict of interest statement in the “Confidential to Editor” section, and submit your "Accept" recommendation.

Reviewer #1: All comments have been addressed

Reviewer #2: All comments have been addressed

2. Is the manuscript technically sound, and do the data support the conclusions?

Reviewer #1: Yes

Reviewer #2: (No Response)

3. Has the statistical analysis been performed appropriately and rigorously? 

Reviewer #1: Yes

Reviewer #2: (No Response)

4. Have the authors made all data underlying the findings in their manuscript fully available?

Reviewer #1: Yes

Reviewer #2: (No Response)

5. Is the manuscript presented in an intelligible fashion and written in standard English?

Reviewer #1: Yes

Reviewer #2: (No Response)

6. Review Comments to the Author

Reviewer #1: (No Response)

Reviewer #2: (No Response)

7. PLOS authors have the option to publish the peer review history of their article (what does this mean? ). If published, this will include your full peer review and any attached files.

**Do you want your identity to be public for this peer review?** For information about this choice, including consent withdrawal, please see our Privacy Policy .

Reviewer #1: **Yes: ** Ludovico Graziani

Reviewer #2: No

---

## [Editor Report · Acceptance letter]

PONE-D-24-57732R1

PLOS ONE

Dear Dr. Holesova,

I'm pleased to inform you that your manuscript has been deemed suitable for publication in PLOS ONE. Congratulations! Your manuscript is now being handed over to our production team.

Kind regards,

on behalf of

Dr. Rishi Jaiswal

Academic Editor

PLOS ONE